# Stabilized Astaxanthin Nanoparticles Developed Using Flash Nanoprecipitation to Improve Oral Bioavailability and Hepatoprotective Effects

**DOI:** 10.3390/pharmaceutics15112562

**Published:** 2023-10-31

**Authors:** Antara Ghosh, Sujan Banik, Kohei Yamada, Shingen Misaka, Robert K. Prud’homme, Hideyuki Sato, Satomi Onoue

**Affiliations:** 1Laboratory of Biopharmacy, School of Pharmaceutical Sciences, University of Shizuoka, 52-1 Yada, Suruga-ku, Shizuoka 422-8526, Japan; antara_pharma007@yahoo.com (A.G.); sbanik@nstu.edu.bd (S.B.); k.yamada@u-shizuoka-ken.ac.jp (K.Y.); h.sato@u-shizuoka-ken.ac.jp (H.S.); 2Department of Bioregulation and Pharmacological Medicine, School of Medicine, Fukushima Medical University, 1 Hikarigaoka, Fukushima 960-1295, Japan; misaka@fmu.ac.jp; 3Department of Chemical and Biological Engineering, A301 EQUAD, Princeton University, Princeton, NJ 80544, USA; prudhomm@princeton.edu

**Keywords:** antioxidative additives, astaxanthin, flash nanoprecipitation, hepatoprotective, nanoparticles, oral bioavailability

## Abstract

In this study, we developed stabilized astaxanthin (AX) nanoparticles (sNP/AX) to improve the physicochemical properties, oral bioavailability, and hepatoprotection of AX. A flash nanoprecipitation technique was used with a multi-inlet vortex mixer to prepare the sNP/AX. Vitamins E (VE) and C (VC) were used as co-stabilizers with poloxamer 407 as a stabilizer to inhibit the oxidative degradation of AX during sNP/AX formation and storage. VC stabilized AX in the aqueous phase during the preparation, whereas VE markedly improved the storage stability of sNP/AX, as evidenced by the AX contents remaining at 94 and 81% after 12 weeks of storage at 4 °C and 25 °C, respectively. The mean sNP/AX diameter was 215 nm, which resulted in higher AX release properties than those of crystalline AX. Rats, orally administered sNP/AX (33.2 mg AX/kg), exhibited higher systemic exposure to AX, whereas oral absorption in the crystalline AX group was negligible. In the rat hepatic injury model, oral pretreatment with sNP/AX (33.2 mg AX/kg) markedly attenuated hepatic damage, as shown by the histopathological analysis and reduced levels of plasma biomarkers for hepatic injury. These findings suggest that strategically including antioxidative additives in the sNP/AX has the potential to improve the physicochemical and nutraceutical properties of AX.

## 1. Introduction

3,3′-Dihydroxy-*β*,*β*-carotene-4,4′-dione (astaxanthin, AX) is a xanthophyll carotenoid with greater potent antioxidant activity than other carotenoids [1]. This compound is extensively distributed in various marine animals and plants such as salmon, shrimp, crab, krill, and microalgae [2,3]. The antioxidant properties of AX are due to its extended polyene structure (Figure 1), which consists of two connected *β*-ionone rings with functional hydroxyl and keto group substitutions. This structure allows for both free-radical scavenging and singlet oxygen quenching activities [4,5]. AX also exhibits a wide range of pharmacological activities, including anti-inflammatory, anticancer, neuroprotective, antihypertensive, and antidiabetic properties [6,7]. However, AX has poor oral bioavailability (BA) due to its extremely low aqueous solubility on account of high lipophilicity (log*P* = 10.3). In addition, AX exhibits poor chemical stability, especially during processing and storage in the presence of light, heat, oxygen, and pH [8,9]. This characteristic limits its application in the food, medicine, and cosmetic industries [10]. Therefore, designing and developing an appropriate formulation system using suitable additives to improve the physicochemical properties and nutraceutical potential of AX would be desirable.

Among the various pharmaceutical approaches, the development of a nanoparticle (NP) formulation is considered the most effective approach for improving the dissolution properties of poorly water-soluble compounds [11,12]. This effect is achieved by generating nano-sized particles with a higher effective surface area for accelerating the dissolution rate. This approach complies with the Noyes–Whitney and Prandtl equation that a decrease in particle size significantly improves the dissolution properties of active substances [13,14]. Rapid precipitation processes can also trap drugs in an amorphous form [15] or a form with nanometer-sized crystals [16], both of which increase solubility dramatically. In this study, flash nanoprecipitation (FNP), an antisolvent precipitation technique, was used to prepare AX NPs. The FNP process, using a multi-inlet vortex mixture (MIVM), has attracted considerable attention for NP fabrication because of its high efficiency of operation, superior controllability, and ease of scalability [17]. MIVM facilitates the rapid and homogenous micromixing of solvent and antisolvent solutions, resulting in a stable NP suspension (nanosuspensions) with amphiphilic polymers used as stabilizers [18,19]. Various amphiphilic block copolymers and random co-polymers can be used as effective stabilizers to produce stable nanosuspensions of highly lipophilic bioactive compounds or drugs (log*P* > 6) with a controlled narrow particle size (<200 nm) [20,21]. AX is highly sensitive to oxygen because of its extended polyene structure. Consequently, AX has been reported to undergoes oxidative degradation during formulation processing, leading to the loss of functional and nutritional properties [22]. Numerous studies have demonstrated that including antioxidative additives significantly improves the physical and chemical stability of AX [23,24,25]. Therefore, in the present study, suitable antioxidative additives, in addition to an amphiphilic block co-polymer, were added during the formulation process to prepare stable AX NPs using FNP.

Considering the above background, in the present study, we sought to design a strategy to improve the physicochemical stability, release kinetics, bioavailability, and nutraceutical properties of AX. Consequently, stabilized AX NPs (sNP/AX) were developed using vitamins E (VE) and C (VC) as antioxidative additives with the amphiphilic block co-polymer poloxamer 407 (P407) as a stabilizer, using FNP. The developed sNP/AX was characterized physicochemically in terms of particle size, morphology, and release properties. A pharmacokinetic study of AX samples was performed in rats to determine whether oral absorption of AX was improved by sNP/AX. Furthermore, the hepatoprotective potential of sNP/AX was evaluated in a rat model of carbon tetrachloride (CCl_4_)-induced acute liver injury.

## 2. Materials and Methods

### 2.1. Materials

The AX extract (33.2% AX, source: *Haematococcus pluvialis*, commercially known as FUJIFILM astaxanthin SS) used to prepare the sNP/AX was purchased from FUJIFILM Co., Ltd. (Tokyo, Japan). Crystalline AX (>96% purity) was obtained from Dr. Ehrenstorfer (Augsburg, Germany). VE and trans-*β*-8′-apo carotenal were purchased from Sigma-Aldrich (St. Louis, MO, USA). P407 and Poloxamer 188 (P188) were received as a kind gift from BASF Japan Co., Ltd. (Tokyo, Japan). Soluplus^®^ and tocopheryl polyethylene glycol 1000 succinate (TPGS) were obtained as generous samples from BASF Japan Co., Ltd. (Tokyo, Japan). VC, CCl_4_, corn oil, dimethyleacetamide, and the liver function assay kit were procured from FUJIFILM Wako Pure Chemical Corporation (Osaka, Japan). All other chemicals and solvents used in the experiment were obtained from commercial sources and were either of analytical or high-performance liquid chromatography (HPLC) grade.

### 2.2. Preparation of sNP/AX

A four-stream MIVM was used to prepare sNP/AX using the FNP process in accordance with previously described methods [17,26]. In FNP process using MIVM, drug solution and antisolvent are injected through four independent inlets, and micromixing of two solvents in MIVM produces high supersaturations, promoting extremely high nucleation rates, and enabling the preparation of fine nanoparticles. An ethanolic solution of the AX extract (drug solution; 5 mL, 20 mg-AX extract/mL with VE 2.5% *w*/*w* AX extract) and an aqueous solution of lecithin (antisolvent; 5 mL, 0.0125% *w*/*v*) were mixed using the MIVM system. The resultant suspensions of NPs were collected in a reservoir containing an aqueous solution (50 mL with P407 2% *w*/*v* and VC 2.5% *w*/*w* AX extract). This nanosuspension was then lyophilized using an Eyela FD-1000 freeze dryer (Tokyo Rikakikai, Tokyo, Japan) to obtain sNP/AX. Trehalose was used as a cryoprotectant during the lyophilization process, and the mass ratio of P407 to trehalose was 2:1.

### 2.3. Ultraperformance LC (UPLC)/Electrospray Ionization (ESI)–Mass Spectrometry (MS) System

The AX concentration in the formulation was quantified using the Waters Acquity ultraperformance LC (UPLC) system (Waters, Milford, MA, USA) equipped with a single quadrupole detector. The assay was conducted with an Acquity UPLC BEH C18 column (2.1 × 50 mm, 1.7 µm particles) and samples (5 µL injection volume) were analyzed using a gradient elution system consisting of Milli-Q (mobile phase A) and acetonitrile (mobile phase B) on the following schedule: 0–2.0 min, 20% A; 2.0–7.5 min, 20–5% A; and 7.5–9 min, 5% A. This was run at a flow rate of 0.25 mL/min and the column temperature was maintained at 40 °C during the analysis, whereas the ion detection was set at m/z 596 for AX [M + H]^+^. This analytical method for the quantification of AX by UPLC-MS system was validated within the range of 5–1000 ng/mL for specificity, linearity (r^2^ = 0.999), accuracy, and precision.

### 2.4. Particle Size Distribution

The mean diameter and polydispersity index (PDI) of the dispersed sNP/AX in Milli-Q water were measured by dynamic light scattering (DLS) using a Malvern Zetasizer Ultra particle sizer (Malvern Instruments Ltd., Worcestershire, UK) at 25 °C with a fixed detection angle of 173°. Briefly, 1 mg of sNP/AX was dispersed in 10 mL of Milli-Q grade water to determine the mean diameter and PDI values of the respective samples. These data were then reported as the means of three (*n* = 3) independent measurements.

### 2.5. Morphology of sNP/AX

The shape and surface morphology of the sNP/AX particles were examined using a Hitachi HT7700 transmission electron microscope (TEM; Hitachi, Tokyo, Japan). An aliquot (5 µL) of dispersed sNP/AX in Milli-Q water was placed on a carbon-coated formvar 200 mesh nickel grid (Nissin EM, Tokyo, Japan) and allowed to stand at room temperature for 10 min. Blotting paper was used to remove the excess suspension from the grid before it was negatively stained with 1% *w*/*v* ammonium molybdate solution and then the sample was visualized to obtain TEM images.

### 2.6. In Vitro Release Study

The release of AX from sNP/AX compared with that of crystalline AX was measured in simulated intestinal fluid (SIF; phosphate buffer [pH 6.8]), containing 0.5% sodium taurocholate (bile salt) as a solubilizer. A dialysis membrane (molecular weight cut off [MWCO] 3.5 kDa, Spectrum Lab, Rancho Dominguez, CA, USA) technique was used according to a previously described study [27]. First, AX samples (3 mg) were dispersed in 1 mL distilled water and placed in dialysis bags, which were immersed in 100 mL release medium at 37 °C while rotating at 100 rpm. Approximately 300 µL of the samples was withdrawn at predetermined time intervals of 1, 2, 3, 5, 8, and 12 h and the same amount of fresh release medium was replaced to maintain the sink condition. The quantity of AX in each sample was measured using the UPLC/ESI-MS system, as described in Section 2.3.

### 2.7. Crystallinity

The crystallinity of AX samples was evaluated through XRPD analysis using a Mini Flex II X-ray diffractometer (Rigaku, Tokyo, Japan) with Cu Kα radiation generated at 40 mA and 35 kV. Data were obtained from 3° to 40° (2*θ*) at a step size of 0.02° and scanning speed of 2°/min.

### 2.8. Stability Studies

The stability of sNP/AX was evaluated by comparing the activities of NPs prepared with and without antioxidants to determine the impact of the antioxidative additives. Each sample (1.0 g) was sealed in an amber glass vial and stored at 4 and 25 °C for 12 weeks. The stored samples were analyzed every 4 weeks for mean diameter, PDI, drug content, and visual appearance. The data obtained at different sampling points were compared to those from freshly prepared samples.

### 2.9. Pharmacokinetic Study

#### 2.9.1. Animals

Male Sprague–Dawley rats (200 ± 5.4 g; 7-week-old; Japan SLC, Shizuoka, Japan) were used in this pharmacokinetic study. Two rats per cage were housed in an environment with consistent access to food and water, a 12 h light/dark cycle, a constant temperature of 24 ± 1 °C, and a relative humidity of 55 ± 5%. All experiments on animals were conducted according to the Animal Facility Regulations of the University of Shizuoka and partially followed Directive 2010/63/EU [28]. The experimental protocols for animal studies were partly aligned with the ARRIVE (Animal Research: Reporting of In Vivo Experiments) Guidelines and received approval from the Animal and Ethics Review Committee of the University of Shizuoka (Approval No. 226548).

#### 2.9.2. Plasma Concentration of AX

Crystalline AX and sNP/AX at a dose of 33.2 mg AX/kg suspended in 1 mL distilled water were orally administered to overnight starved rats. To calculate the oral BA of the AX samples, AX solution was dissolved in dimethylacetamide and polyethylene glycol 40 (50:50 *v*/*v*), and a dose of 1 mg/kg was administered intravenously (iv) to rats for the calculation of absolute BA [29]. Then, blood samples (approximately 400 µL) were collected through the tail veins of rats into heparinized tubes at predetermined time points. These samples were then centrifuged at 6000× *g* for 10 min to obtain plasma samples. The plasma samples were stored at −80 °C until further analysis.

Deproteinization of plasma samples (100 µL) was accomplished by adding 300 µL acetone containing trans-*β*-apo-8′-carotenal (6.7 µg/mL) as an internal standard, vortexed for 1 min, and then centrifuged at 10,000× *g* for 10 min at 4 °C. To de-esterify the AX, the supernatants were collected and treated for 2 h with an enzymatic solution of cholesterol esterase (1 mL). The collected samples were then evaporated at 37 °C under nitrogen gas flow, the residues dissolved in 100 µL acetonitrile, and were filtered through a 0.2 µm membrane filter (Millex LG, Millipore, Billerica, MA, USA). Finally, the plasma AX concentration was determined using the UPLC/ESI-MS system, as outlined in Section 2.3.

### 2.10. Evaluation of Hepatoprotective Effects

#### 2.10.1. Rat Model of Acute Hepatic Injury

The hepatoprotective effects of AX samples were evaluated using a rat model of acute hepatic injury induced using CCl_4_ [30]. Male SD rats, weighing 180–200 g (7 weeks of age), were randomly assigned into four groups; each group consisted of four rats, as follows: group I: control group (untreated rats); group II: vehicle group; group III: crystalline AX treatment group; and group IV: sNP/AX treatment group. Except for the rats in group 1, all of the rats were administered CCl_4_ (0.7 mL/kg body weight) orally along with an equal volume of corn oil to induce acute hepatic injury. As a control experiment, rats in group I were given corn oil at a dose of 1.4 mL/kg body weight. Crystalline AX and sNP/AX (33.2 mg AX/kg body weight) dispersed in 1 mL of distilled water were administered orally to rats in groups III and IV 3 h before the CCl_4_ treatment.

To measure the plasma biomarker levels of the hepatic injury, approximately 300 µL of blood samples were collected from the tail vein of each rat 3, 6, 12, and 24 h after the CCl_4_ treatment. The collected blood samples were centrifuged at 10,000× *g* for 10 min to obtain plasma samples, which were stored at −80 °C until analysis. The rats were then euthanized to collect liver tissue samples, which were rinsed in ice-cold saline and blotted with filter paper.

#### 2.10.2. Histopathological Observation

The liver tissues were promptly fixed in 10% neutral buffered formalin. Following three washes with phosphate-buffered saline (PBS, pH 7.4), the samples were immersed for 24 h at 4 °C in a 30% sucrose solution with 0.1% sodium azide. The tissues were then put into an optimal cutting temperature (OCT) compound (Sakura Finetek, Tokyo, Japan), frozen using liquid nitrogen, and cut into 10 µm thick slices using a CM1520 cryostat (Leica Microsystems, Tokyo, Japan). The liver sections were then stained with hematoxylin and eosin (H&E) to examine the histopathological changes using a light microscope.

#### 2.10.3. Plasma Biomarkers Level

Plasma levels of alanine aminotransferase (ALT) and aspartate aminotransferase (AST) were quantified as biomarkers for liver damage utilizing a test kit obtained from FUJIFILM Wako Pure Chemical Corporation (Osaka, Japan). The spectrophotometric determination of ALT and AST plasma levels was measured at a wavelength of 340 nm using a microplate reader, Safire (Tecan, Männedrof, Switzerland). Each sample was evaluated in triplicate.

#### 2.10.4. Hepatic Tissue Distribution

To measure the AX concentration in the liver tissue, samples (1.0 g) were homogenized in 4 mL of PBS (pH 7.4) with 0.005% VC using a Physcotron homogenizer (Microtec Co., Ltd., Chiba, Japan). The homogenates were transferred into stoppered test tubes and mixed with the same volume of chloroform containing 0.005% VE and an internal standard (trans-*β*-apo-8′-carotenal, 6.7 µg/mL). The isolated organic phase, after centrifugation at 2000 rpm for 5 min, was transferred into a glass tube and evaporated under a nitrogen gas stream at 37 °C. The resulted residues were dissolved in 100 µL acetonitrile, filtered through a 0.2 µm membrane filter, and subsequently subjected to AX concentration measurement using the UPLC/ESI-MS system, as explained in Section 2.3.

### 2.11. Statistical Analysis

All experiments were performed at least in triplicate, and the values are expressed as mean ± standard deviation or mean ± standard error. GraphPad Prism software (version 8.0.1, GraphPad, San Diego, CA, USA) was used for statistical analysis using a one-way analysis of variance followed by Tukey’s multiple comparisons test, and a *p*-value of less than 0.05 was considered to be statistically significant.

## 3. Results and Discussion

### 3.1. Development of sNP/AX

#### 3.1.1. Selection of Amphiphilic Block Co-Polymer as Stabilizer

In the present study, sNP/AX were developed using an FNP process with an MIVM. MIVM is a microfluidic mixing device that can produce NPs through the rapid and homogeneous mixing of solvents and anti-solvents from the four inlets with a narrow size distribution. Particle size reduction using a bottom-up approach with the FNP technique is one of the most efficient approaches for improving the dissolution properties and delivery of poorly water-soluble compounds [11]. This technique is also highly advantageous for thermolabile compounds like AX because it does not require a large energy input or heat production. In addition, polymers or surfactants are also simultaneously added to stabilize the NPs to prevent aggregation [20,21]. To choose a suitable polymer as a stabilizer, several amphiphilic block copolymers, including Soluplus^®^, TPGS, P188, and P407, were evaluated in this study. P407 was subsequently selected as the most suitable stabilizer because it provided the highest recovery of AX from nanosuspensions and prevented the NPs from aggregation (Table 1). The large hydrophilic and hydrophobic blocks of P407 possibly contributed to stabilizing the NPs by strengthening the adsorption of AX on the hydrophobic block or encapsulating AX into the hydrophobic core of the polymer. Furthermore, hydrophilic blocks dispersed throughout the surface of the NPs provide stearic stabilization and prevent aggregation. Moreover, P407 is industrially available and is a safe excipient because of its low toxicity after oral administration based on its median lethal dose (LD_50_) value of >5000 mg/kg in rats [31].

#### 3.1.2. Stabilizing Effects of Antioxidative Additives during Preparation and Storage

The highest recovery of AX from the nanosuspensions after the FNP process without co-stabilizers was found to be approximately 45% after freeze-drying. Furthermore, this value significantly decreased to 11% and below the detection limit after 4 weeks of storage at 4 and 25 °C, respectively (Table 2), possibly due to the degradation of AX during preparation and storage. Extensive studies on carotenoid stability under various conditions have reported their instability in the presence of light, heat, and oxygen, which is due to their polyene conjugated chemical structure [32,33,34]. The MIVM process was used to make the NPs without employing high temperatures or exposure to light. However, the presence of dissolved oxygen in the aqueous phase of the nanosuspensions might have accelerated the oxidation of AX during the FNP.

Previous studies have reported that atmospheric oxygen may cleave the double bond of AX, causing epoxidation and hydroxylation [35]. These effects lead to color changes in AX, and auto-oxidation and first-order kinetics of degradation were observed during storage [36]. Therefore, the antioxidative additives VE and VC were used as co-stabilizers with P407 to inhibit the oxidative degradation of AX during NP preparation and processing. Furthermore, the ratio of these agents was optimized based on AX recovery after the freeze-drying and then the product was stored under 4 and 25 °C conditions for 4 weeks (Table 2). The results showed that using VC alone with P407 slightly improved the recovery of AX but could not maintain the residual content for 4 weeks of storage at 4 and 25 °C. In contrast, combining VE with P407 did not improve the recovery of AX after the preparation process, but it maintained the AX content under both storage conditions.

Using VE and VC together significantly improved the AX content retention and stability during storage, as evidenced by the AX content in the sNP/AX remaining at approximately 94 and 82% at 4 and 25 °C, respectively, after 4 weeks of storage (Table 2). The AX recovery results suggested that combining VE (2.5%) and VC (2.5%) with P407 was appropriate for developing the sNP/AX. This result suggests that adding VC to the aqueous phase during the FNP process inhibited the oxidative degradation of AX caused by dissolved oxygen. In addition, the VE in the preparation enhanced the stability of AX during storage under solid state owing to its close contact with AX molecules. The stability analysis of the AX samples after 12 weeks of storage at 4 and 25 °C (Appendix A) revealed that the AX NPs without antioxidative additives showed significant color changes, with an almost complete discoloration to white at 4 and 25 °C. In contrast, the sNP/AX maintained their initial color even after 12 weeks of storage at 4 °C and showed very slight color changes when stored at 25 °C for 12 weeks. Moreover, the AX contents of the sNP/AX, even after 12 weeks of storage at 4 and 25 °C, were 94 and 81%, respectively, suggesting their high stability under both conditions. Therefore, the inclusion of antioxidative additives could potentially improve the chemical stability of sNP/AX in stressful environments.

### 3.2. Physicochemical Characterization of sNP/AX

The mean particle sizes (diameter) of the freshly prepared and freeze-dried sNP/AX were 205 ± 1.05 and 215 ± 6.18 nm, respectively, with PDI values of 0.15 ± 0.01 and 0.31 ± 0.02, respectively. These results suggest good dispersibility in distilled water with a narrow particle size distribution (Figure 2A). After 12 weeks of storage at 4 and 25 °C, the mean particle size of the sNP/AX was similar to that of the fresh samples, demonstrating good stability during storage. In addition, the TEM analysis showed that the NPs dispersed in distilled water had a uniform and spherical shape without significant aggregation (Figure 2B). The particle sizes observed in the TEM analysis were slightly different from those in the DLS analysis. This discrepancy is most likely attributable to the differences between the two techniques. DLS measures a hydrodynamic size where the hydrated corona of the stabilizing polymer makes a substantial contribution to the measured diameter, whereas TEM analysis is conducted on dry samples where the corona is compressed on the NP surface. These findings suggest that the sNP/AX had a relatively narrow size distribution and good dispersibility, which might contribute to improving the dissolution properties and oral absorption of AX.

### 3.3. Release Properties of sNP/AX

The comparison of the release properties of AX is shown in Figure 3. The results showed that even with a solubilizing agent in the dissolution media, the amount of AX released from crystalline AX through the dialysis membrane was below the detection limit (˂5 ng/mL). This was due to its extremely low aqueous solubility and dispersibility. In contrast, sNP/AX exhibited a significant improvement in the release rate of AX through the dialysis membrane. This effect can be attributed to the enhanced dissolution properties of sNP/AX owing to its reduced particle size and the partial amorphization of AX in sNP/AX (Appendix A) [37]. In addition, the sustained-release behavior was most probably achieved by the stable incorporation of highly lipophilic AX (log*P* = 10.3) into the micellar core and slow diffusion of AX from the micelles composed of P407.

Hydrophobic drugs or bioactive compounds have been reported to exhibit zero-order release profiles, particularly when encapsulated in polymeric NPs [38]. Overall, the present study findings imply that sNP/AX could be an ideal delivery system for the enhancement of the oral absorption of AX.

### 3.4. Pharmacokinetic Studies in Rats

The results of the pharmacokinetic studies in rats including the plasma-concentration time profile of AX and the respective pharmacokinetic parameters are presented in Figure 4 and Table 3, respectively. As shown in Figure 4A, the oral absorption of AX from crystalline AX was negligible at every time point and below the limit of detection (5 ng/mL) owing to its poor aqueous solubility and high lipophilicity. In contrast, orally administered sNP/AX exhibited an improved systemic release of AX with calculated maximum concentration (*C*_max_) and area under the curve from time 0 to 24 h (AUC_0–24 h_) values of 17 ± 3.5 ng/mL and 298 ± 12 ng·h/mL, respectively, indicating significant improvement in the oral BA of AX. The absolute oral BA of sNP/AX was calculated to be 2.2% based on the AUC_0−inf_. value of an intravenously administered AX solution (1.0 mg/mL, 400 ± 54 ng·h/mL) (Figure 4B). The delayed time to achieve *C*_max_ (*T*_max_, 3.0 h) indicated a relatively sustained release of AX from sNP/AX, which agreed with the results of the release studies. The mechanism of carotenoid absorption followed by solubilization in bile acid micelles in the intestinal tract was reported in a previous study [39]. A previous study also reported that particle sizes < 300 nm with good dispersibility could be easily absorbed by enterocytes, which increased the oral absorption [40].

In the present study, sNP/AX might have facilitated the absorption of AX through enterocytes by improving its dissolution in the gastrointestinal tract and reducing the thickness of the diffusion layer. Moreover, P407, used as a stabilizer in the sNP/AX, might partially accelerate the permeation of AX through enterocytes, because amphiphilic polymers act as a permeability enhancer [41]. To date, very few studies have reported the absolute oral BA of AX, and only the enhancement of its pharmacokinetic parameters has been reported [42,43]. In the present study, the results revealed that stabilizing AX NPs using antioxidative additives is effective for enhancing the oral BA of AX. Therefore, the enhancement of the oral absorption of AX using sNP/AX could potentially improve its beneficial effects for the treatment of free-radical mediated diseases such as liver injury or fibrosis.

### 3.5. Hepatoprotective Effects of sNP/AX

AX is a potent bioactive antioxidant that breaks lipid peroxidation chains by scavenging highly reactive oxygen species [44]. AX has been shown to increase hepatic endogenous antioxidant enzymes including superoxide dismutase and catalase, which might help in preventing free radical-induced oxidative disorders [45]. In this study, sNP/AX improved the oral absorbability of AX and, hence, sNP/AX may exhibit improved hepatoprotective effects against CCl_4_-induced hepatic injury in rats.

The results of the histological analysis of H&E-stained liver tissues, illustrated in Figure 5A, showed that the control group had well-defined cellular structures with healthy cytoplasm and hepatic cells. In contrast, the liver tissues of the CCl_4_-treated group showed visible damage with significant fatty changes (hepatic steatosis), fibrotic spectra between the nodules, necrosis, and the degeneration of parenchymal cells (Figure 5B). CCl_4_ is commonly used as a hepatotoxin to establish experimental rat models of acute liver injury [46].

CCl_4_ can be metabolized in the liver by cytochrome P450 (CYP)-dependent enzymes into a reactive trichloromethyl radical (•CCl_3_), which can be further converted to the more destructive trichloromethyl peroxyl radical (CCl_3_OO•). These radicals react with oxygen and react with cellular macromolecules including lipids, proteins, and nucleic acids, causing free radical-mediated lipid peroxidation, leading to hepatic damage [47]. The group treated with sNP/AX (33.2 mg AX/kg, orally) showed a marked improvement in the hepatic steatosis and retention of a normal hepatocellular structure with reduced necrosis (Figure 5D). In contrast, oral pretreatment with crystalline AX (33.2 mg/kg) did not maintain the normal hepatocellular structure (Figure 5C).

In this study, plasma ALT and AST levels were estimated as surrogate biomarkers of hepatotoxicity to investigate the hepatocellular damage in the CCl_4_-treated rat model [48]. The results showed that the levels of both ALT and AST were significantly higher in rats treated with CCl_4_ alone than they were in the control group, indicating substantial injury to hepatic tissues (Figure 6). In contrast, pretreatment with sNP/AX significantly reduced the elevated plasma ALT and AST levels by 62 and 51% (*p* < 0.01), respectively, compared with levels of the vehicle group.

Furthermore, pretreatment with crystalline AX showed no significant difference in ALT and AST levels compared with the levels of the vehicle-treated group, owing to the negligible oral absorption of crystalline AX (Figure 4). As shown in Figure 7, the AX concentrations in hepatic tissues after oral administration of sNP/AX (33.2 mg AX/kg) were 476.53 ± 61 ng/g tissue, whereas those following a similar dose of crystalline AX were below the detection limit (5 ng/mL). These findings suggest that the enhanced hepatic biodistribution of AX by sNP/AX contributed to improving the hepatoprotective potential of AX. AX supplementation at the dose of 80 mg/kg per day exhibited a significant protective effect against liver fibrosis [49]. In addition, AX attenuated liver injury in concanavalin A-induced autoimmune hepatitis by inhibiting the release of inflammatory factors such as tumor necrosis factor (TNF)-α, interleukin (IL)-6, IL-1β, and interferon (IFN)-γ [50]. Therefore, sNP/AX could attenuate hepatocellular damage and would be beneficial for preventing free-radical mediated disorders.

## 4. Conclusions

The sNP/AX developed in this study, including antioxidative additives using an FNP process, was found to be highly stable during preparation, processing, and storage. This formulation exhibited a sustained release of AX, which significantly enhanced its oral BA compared with that of crystalline AX. Additionally, sNP/AX markedly enhanced the hepatoprotective effects of AX in a rat model of acute liver injury. These findings indicate that the strategic inclusion of antioxidative additives to sNP/AX has the potential to improve the physicochemical stability and nutraceutical properties of AX and other oxygen-labile compounds.

## Figures and Tables

**Figure 1 pharmaceutics-15-02562-f001:**
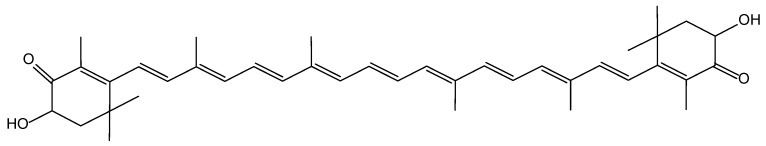
Chemical structure of astaxanthin (AX).

**Figure 2 pharmaceutics-15-02562-f002:**
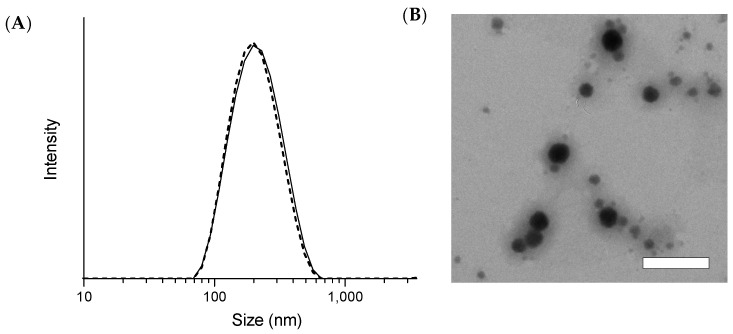
(**A**) Particle size distribution of stabilized astaxanthin nanoparticles (sNP/AX) was measured using dynamic light scattering (DLS) before and after freeze-drying. Solid and dashed lines indicate particle size distribution before and after freeze-drying, respectively. (**B**) Transmission electron microscopy (TEM) analysis of dispersed sNP/AX in distilled water. White bar represents 200 nm.

**Figure 3 pharmaceutics-15-02562-f003:**
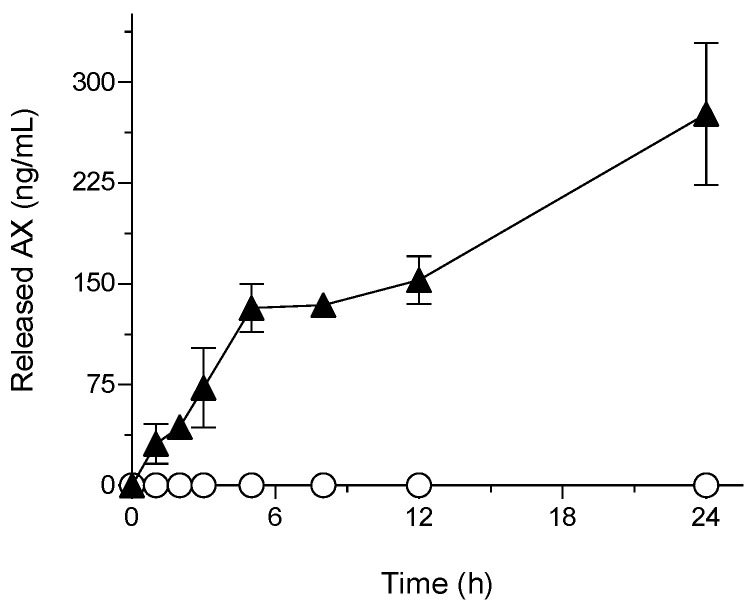
Astaxanthin (AX) release profile in simulated intestinal fluid (pH 6.8). ○, crystalline AX; and ▲, sNP/AX. Data represent mean ± standard deviation (*n* = 3).

**Figure 4 pharmaceutics-15-02562-f004:**
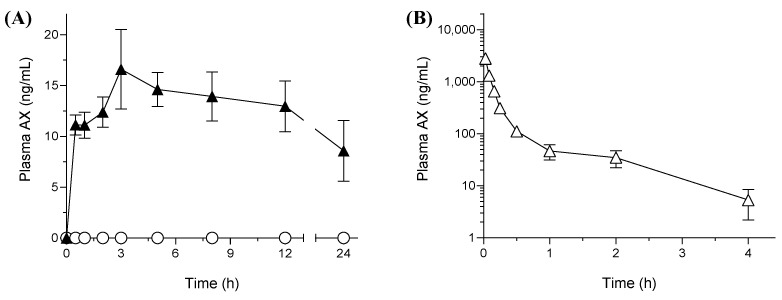
Plasma concentration profile of astaxanthin (AX) after oral administration (**A**) of AX samples (33.2 mg AX/kg) and intravenous administration (**B**) of AX (1 mg-AX/kg) to rats. ○, crystalline AX, and ▲, sNP/AX. Data are mean ± standard error (*n* = 4).

**Figure 5 pharmaceutics-15-02562-f005:**
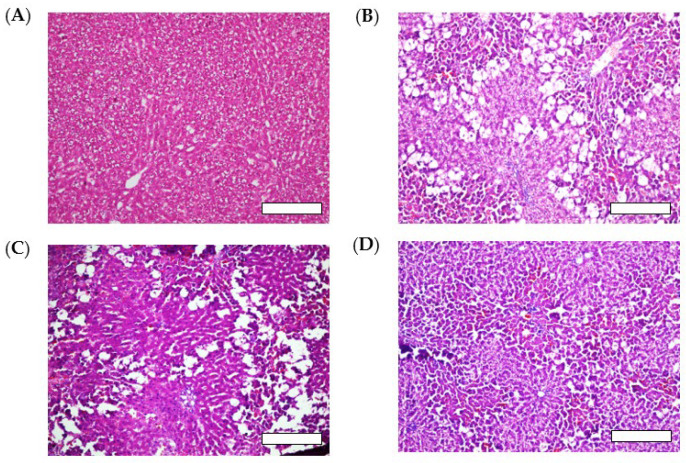
Histological evaluation of liver tissues after carbon tetrachloride (CCl_4_) challenge with or without oral administration of AX samples (33.2 mg AX/kg). Rats treated with (**A**) corn oil (control), (**B**) CCl_4_ (vehicle), (**C**) CCl_4_ with crystalline AX, and (**D**) CCl_4_ with sNP/AX. Each bar represents 100 µm.

**Figure 6 pharmaceutics-15-02562-f006:**
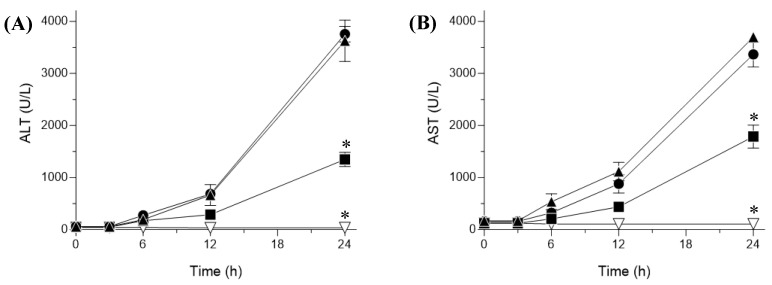
Plasma biomarker levels indicating hepatic injury in carbon tetrachloride (CCl_4_)-challenged with or without pretreatment with astaxanthin (AX) samples. Plasma (**A**) ala-nine aminotransferase (ALT) and (**B**) aspartate aminotransferase (AST) levels. ▽, control (corn oil-treated rats); ▲, vehicle (CCl4-challenged rats); ●, CCl4-challenged rats treated with crystalline AX; and ■, CCl4-treated rats with sNP/AX. Data represent mean ± standard error (*n* = 6). * *p* < 0.01 compared with vehicle.

**Figure 7 pharmaceutics-15-02562-f007:**
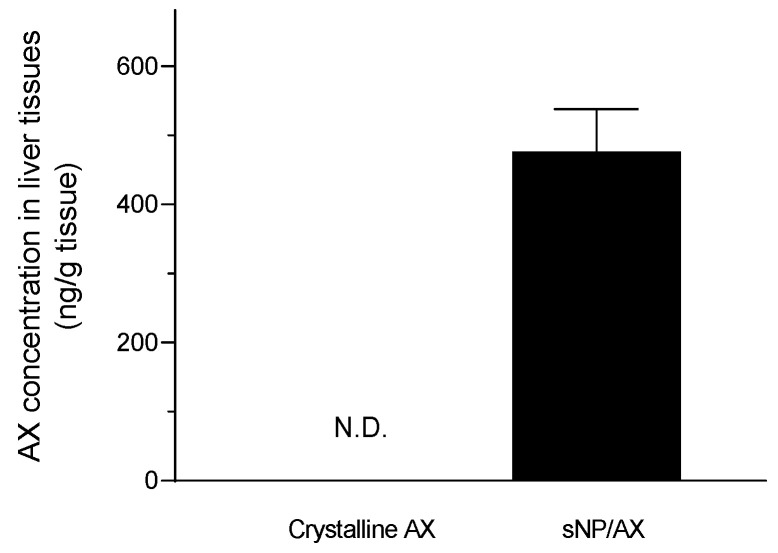
Hepatic tissue distribution of astaxanthin (AX) following oral administration of AX samples (33.2 mg AX/kg). N.D., no detection. Data represent mean ± standard error (*n* = 4).

**Table 1 pharmaceutics-15-02562-t001:** Selection of suitable stabilizer to prevent nanoparticles aggregation.

Stabilizer	Particle Size (nm)	PDI	AX Recovery (%)
P188	186 ± 2.6	0.19 ± 0.01	73 ± 0.6
P407	187 ± 1.5	0.14 ± 0.03	89 ± 7.3
TPGS	170 ± 6.5	0.38 ± 0.03	82 ± 6.2
Soluplus^®^	155 ± 1.5	0.19 ± 0.01	68 ± 3.5

P188, poloxamer 188; P407, poloxamer 407; TPGS, tocopheryl polyethylene glycol 1000 succinate; AX, astaxanthin. Data represent mean ± standard deviation (*n* = 3).

**Table 2 pharmaceutics-15-02562-t002:** Effects of antioxidative additives vitamins E (VE) and C (VC) on properties of stabilized astaxanthin nanoparticles (sNP/AX).

Stabilizers	Initial Samples		Stored Samples (4 Weeks, 4 °C)		Stored Samples (4 Weeks, 25 °C)	
	Particle Size (nm)	AX Recovery (%)	Particle Size (nm)	Remaining AX (%)	Particle Size (nm)	Remaining AX (%)
P407	236 ± 3.6	45	278 ± 12	11	289 ± 15	N.D.
P407 + VC (2.5%)	289 ± 7.8	54	-	-	-	-
P407 + VC (5.0%)	266 ± 5.5	54	270 ± 50	7	299 ± 45	N.D.
P407 + VE (2.5%)	260 ± 5.0	46	-	-	-	-
P407 + VE (5.0%)	255 ± 5.5	40	270 ± 45	34	266 ± 49	34
P407 + VE (2.5%) + VC (2.5%)	215 ± 6.8	>99	212 ± 12	94	242 ± 13	82

P407, poloxamer 407; N.D., not detected. Data represent mean ± standard deviation (*n* = 3).

**Table 3 pharmaceutics-15-02562-t003:** Pharmacokinetic parameters of astaxanthin (AX) after oral administration of AX samples (33.2 mg AX/kg) to rats.

Parameters	Crystalline AX	sNP/AX
*C*_max_ (ng/mL)	<LOD	17 ± 3.5
*T*_max_ (h)	<LOD	3.0 ± 0
AUC_0–24 h_ (ng·h/mL)	–	298 ± 12
BA (%)	–	2.2

*C*_max_, maximum concentration; *T*_max_, time to maximum concentration; AUC_0–24 h_, area under the curve of plasma concentration vs. time from 0 to 24 h; BA, absolute bioavailability; LOD, limit of detection (5 ng/mL). Data represent mean ± standard error (*n* = 4).

## Data Availability

Not applicable.

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
