# Peer review of "Stabilized Astaxanthin Nanoparticles Developed Using Flash Nanoprecipitation to Improve Oral Bioavailability and Hepatoprotective Effects"

_pharmaceutics, 2023, doi:10.3390/pharmaceutics15112562_

Round 1
Reviewer 1 Report
Comments and Suggestions for Authors
Manuscript ID: pharmaceutics-2650403
Title: Stabilized astaxanthin nanoparticles developed using flash nanoprecipitation to improve oral bioavailability and hepatoprotective effects
Author: Antara Ghosh, Sujan Banik, Kohei Yamada, Shingen Misaka, Robert K. Prud’homme, Hideyuki Sato and Satomi Onoue
Overview and general recommendation:
The study focused on developing stabilized astaxanthin (AX) nanoparticles (sNP/AX) to improve the physicochemical properties, oral bioavailability, and hepatoprotection of AX using the flash nanoprecipitation technique. The aim of the study is precise either in the abstract or the introduction. The study is well-designed and written, and the methodology is well mentioned with the required detail. The results are well discussed with good visualization.
However, I ask the authors to address the recommendations for improving the manuscript before acceptance for publication.
Comments to the authors:
1) On line 79, what does the authors mean with “A pharmacokinetic study of AX release”? Is it a PK or a release?
2) It is recommended that some details in the preparation of astaxanthin nanoparticles despite of the cited reference for reproducibility purposes.
3) Figure 2, the scale bar in the TEM image is not precise. The particle size of the formulation was larger than 200 nm as mentioned in Table 2 and Figure 2A while the size in TEM is smaller than 100 nm, as declared from the scale bar. Please justify.
4) On line 344, the authors stated that significant improvement in the release rate of AX can be attributed to the enhanced dissolution properties of sNP/AX owing to its reduced particle size and increased solubility of the amorphous AX. I agree with the authors' justification regarding the reduction of particle size confirmed during the study, whereas the amorphous structure of AX is not confirmed. Please clarify.
5) In Table 3, How do the authors calculate the BA while there is no numerical value of the AUC of the crystalline drug?
Comments on the Quality of English LanguageMinor editing of English language required.
Author Response
※Please see uploaded PDF file.

Reviewer 2 Report
Comments and Suggestions for Authors
"Stabilized astaxanthin nanoparticles developed using flash nanoprecipitation to improve oral bioavailability and hepatoprotective effects" is an interesting paper with experimental designs showing improved bioavailability and efficacy studies too. I have minor queries,
1) Can the authors exactly identify the degree of crystallinity of pure drug and that after nanoparticles?
2) Since VC was added to improve the stability of the drug against oxidation. How was the concentration of VC optimized?
Author Response
※Please see uploaded PDF file.

Round 2
Reviewer 1 Report
Comments and Suggestions for Authors
The authors have addressed my concerns. However, I recommend the combination of the plasma concentration time curve after intravenous administration with the oral curve in the manuscript (not in the supplementary files). Otherwise, the manuscript has been improved and can be accepted for publication.
Author Response
Please see uploaded PDF file.
